# Numerical study on the stress and deformation behavior of buried PE pipeline exposed to stacking load

**Pengcheng Li[1], Dong Lin[1], Ming Wen[1], Fei Wang[2], Chang Liu[1], Mingnan Sun[1], Jianguo Lu[3]***

1 Institute of Safety, Environment Protection and Technical Supervision, PetroChina Southwest Oil & Gas Field Company, Chengdu, China, 2 Pipeline Management Department, PetroChina Southwest Oil & Gas Field Company, Chengdu, China, 3 School of Civil Engineering and Geomatics, Southwest Petroleum University, Chengdu, China

* jianguog@swpu.edu.cn

## Abstract

Oil and gas pipelines serve as critical infrastructure for ensuring energy resource supply and security. Stacking load, as one of the most widespread loads, frequently acts on the soil surface overlying buried polyethylene (PE) pipelines, influencing stress-strain dynamics and potential failure mechanisms. In this study, a numerical simulation model of pipeline-soil interaction for buried PE pipelines exposed to stacking load was established and verified. The variations in deformation and stress behavior were systematically analyzed. The results revealed that the maximum deformation of buried PE pipelines under stacking load occurred at the pipeline's center, with upper and lower deformations, as well as diameter variations, diminishing progressively as the distance from the center increased. Furthermore, the difference between the upper and lower deformations was reduced as burial depth increased. Notably, the maximum deformation exhibited a direct proportionality with the stacking load, while both the deformation and its rate of reduction decreased with buried depth increased. Additionally, the peak stress within the buried PE pipeline emerged at the center of the load's footprint, escalating in tandem with the increase in stacking load. Intriguingly, the stress response diverged between pipelines subjected to smaller and larger stacking loads, and the peak stress gradually attenuated as the pipeline's diameter expanded. These findings highlight the importance of considering both stacking loads and buried depths in the design and installation for PE pipelines to safeguard structural integrity and extend operational lifespan.

## 1 Introduction

By the end of 2020, China's comprehensive network of oil and gas pipelines had attained a total mileage of 14.5 million kilometers, with natural, crude oil, and refined

---

**Data availability statement:** All relevant data are within the manuscript and its Supporting Information files.

**Funding:** This study was supported by the National Key R &D Program of China (2024YFC3810702), Science and Technology Project of PetroChina Southwest Oil and Gas Field Company (Grant No. 20220307-10), and the Natural Science Foundation of Sichuan Province (Grant No. 2024NSFSC2007). The specific roles of this author are articulated in the 'author contributions' section. The funders had no role in study design, data collection and analysis, decision to publish, or preparation of the manuscript.

**Competing interests:** The authors have declared that no competing interests exist.

oil products pipelines constituting approximately 8.54 million kilometers, 3.05 million kilometers, and 2.91 million kilometers, respectively. Oil and gas pipelines play a pivotal role in sustaining and safeguarding China's oil and gas resources, as well as energy security [1–3]. As the proportion of natural gas in China's energy mix continues to grow, polyethylene (PE) has emerged as a widely favored material for urban buried gas pipelines due to its inherent advantages, including exceptional corrosion resistance, minimal flow resistance, superior seismic stability, outstanding mechanical properties, and cost-effectiveness. Recently, due to large-scale urban construction and renovation, over-exploitation of groundwater resources, and increasing number of transportation facilities, the original mechanical equilibrium of the natural soil layer is damaged and the ground subsidence rate is aggravated. Soils in proximity to the settlement area can experience subsidence or slipping along fracture surfaces, subjecting the buried gas pipeline to significant normal pressure and tangential shear forces. These forces can potentially lead to damage to the pipeline, posing a risk to its integrity and operational safety [4]. Crude oil, refined oil, and natural gas transported by pipelines are flammable and explosive. Once the pipeline failure leakage occurs, fires and explosions accidents might be occurred, which will cause catastrophic impact on the safety of human life and natural environment in the densely populated or environmentally sensitive high-consequence areas [5]. Among them, the form of pipeline strain resulting in cracks or yield failures includes but is not limited to soil subsidence caused by environmental problems (e.g., freeze-thaw, dry-wet) [6,7], overexploitation of groundwater [8,9], mining [10], tunnel engineering [11] and loose foundations [12].

To address the issue of pipeline damage caused by multiple factors, researchers have employed numerical simulations and laboratory tests to examine the stress and strain characteristics of pipelines [6,13,14]. Many scholars regarded pipelines as beams on an elastic basis, or solve pipeline stress problems based on a more rigorous elastic continuum formula [11,15,16]. Xu et al. [17] established a coupling model for pipeline-soil-oil interactions and discovered that the presence of oil within the pipeline induced additional deformation effects. Through numerical simulations of PE pipelines subjected to foundation settlement, the maximum stress in PE pipelines was associated with the settlement displacement, and critical fractures were occurred in the transition sections at the junctions between the settled and non-settled areas [18]. Miad et al. [19] carried out a numerical simulation of pipeline-soil interaction behavior using a beam-spring-interface unit, and compared the simulation results with experimental values to validate the proposed method, confirming the method's accuracy in a 3D finite element analysis. Considering the effects of the buried depth, reinforcement material, and cyclic loading on the overall system, Ahmed et al. [20] conducted 3D numerical simulations of a reinforced soil pipelines system subjected to cyclic loading. Their findings indicated that incorporating geogrids facilitates the redistribution of pressure within the rigid layer and enhances lateral support for the pipeline. Rajeev and Kodikara [7] carried out experimental and numerical analyses to quantify pipeline-soil interaction effects in expansive soils.

Moreover, large number of numerical simulation studies have been carried out to accurately predict the mechanisms of action and the forms of damage to buried pipelines

[21–24]. Ma et al. [25] conducted a thorough study examining the interplay between buried pipelines and soils, specifically focusing on the effects of temperature and water supply conditions. Their findings revealed that when the ambient temperature dropped below −10°C, the influence of water supply conditions on the maximum frost heave phenomenon significantly diminished. This insight offered valuable information for understanding and mitigating potential pipeline-soil interactions under extreme temperature conditions. Fang et al. [26] explored the influential factors, such as depth and width of corrosion, traffic load, and cover depth, on the mechanical properties of buried pipelines. To investigate the peak effect during pipeline installation, the effects of soil compaction and vertical cover on the deformation of buried pipelines were examined, and concluded that stress decreased as the external load applied above the pipeline [27]. Xie et al. [28] employed the material point method to investigate the bearing capacity and damage mechanisms of pipeline-soil structures, and elucidated the variations in damage modes associated with pipeline-soil interactions and provided novel theoretical support for the construction of buried pipeline systems. Besides, the mechanical properties of buried PE pipelines under extreme load conditions were also studied [29,30]. These studies provide valuable insights and serve as a reference for analyzing incidents involving buried PE pipelines. However, numerical simulation is insufficient for accurately observing the operation of buried pipelines under external loads.

Currently, there is limited research on the stress-deformation coupling mechanism of buried PE pipelines under varying loads and burial depth conditions. This study establishes a pipeline-soil coupling model and derives empirical formulas for pipeline stress in relation to burial depth, as well as pipeline displacement in relation to stacking load. The findings provide an important theoretical basis and practical guidance for the safe construction and operation of pipelines. In this study, a calculated model of pipeline-soil interaction for buried PE pipelines exposed to stacking load was established. Then, the variations in deformation and stress of buried PE pipeline were analyzed, and the influential factors on the deformation and stress were discussed. This study could provide useful references for understanding the variations of deformation and stress for buried PE pipeline exposed to stacking load, as well the reliability and safe operations of buried PE pipelines under complicated working conditions.

## 2. Model description

### 2.1 Governing equations

The stress action between the buried PE pipeline and foundation soils is a dynamic equilibrium process. The stacking load will change the stress transfer and dissipation characteristics of foundation soils, and significantly impact the stress redistribution of buried PE pipeline, resulting the interaction process between the buried PE pipeline and foundation soils more complicated. Considering the deformation coordination conditions of the buried PE pipeline and foundation soils, a buried PE pipeline-foundation soils coupling calculation model exposed to stacking load is established.

The buried PE pipeline is simplified as a foundation beam, assuming that the longitudinal stiffness of the pipeline is constant, only rigid movement and rotation are generated, and the foundation reaction at any point is proportional to the foundation subsidence at the point. Thus, only the settlement within the load range is considered. Fig 1 shows the schematic diagram of calculation model between the buried PE pipeline and foundation soils, the left end of the PE pipeline (at the axis) is defined as the origin, and the direction of PE pipeline axis is the x-axis.

In the simulation calculation, the uniform distribution load is applied on the soil surface, the coordinates of the load center is O $(x_p, y_p, z_p)$, the length and width of applied load are 2b and 2a, respectively (Fig 1). The foundation is a semi-infinite elastic body, and the vertical and lateral forces of the buried PE pipeline can be calculated according to Eqs. 1 and 2, respectively.

$$q_v = -\frac{3pz^3}{2\pi} \int_{y_p-b}^{y_p+b} \int_{x_p-b}^{x_p+b} \frac{1}{R^5} dxdy$$

(1)

$$q_u = -\frac{3p}{2\pi} \int_{y_p-b}^{y_p+b} \int_{x_p-b}^{x_p+a} \left\{ \frac{y^2z}{R^5} + \frac{1-2\mu}{3} \left[ \frac{R^2-z(R+z)}{R^3(R+z)} - \frac{y^2(2R+z)}{R^3(R+z)^2} \right] \right\} dxdy$$

(2)

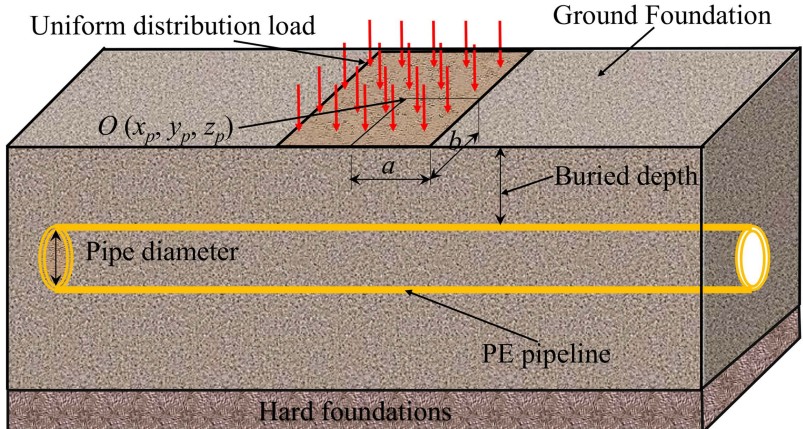

**Fig 1. Schematic diagram of calculation model between the buried PE pipeline and foundation soils.**

where $p$ is the uniform distribution load, $R$ is the distance between the calculated point and load center in model, it can be obtained as,

$$R^2 = (x-\tau)^2 + y^2 + z^2 \tag{3}$$

where $\tau$ is the coordinate value in the x-direction.

The settlement of soils in the foundation can be calculated by layer-summation method,

$$\delta(x) = -\frac{3\rho}{2E\pi} \int_{z_1}^{z_2} \int_{y_p-b}^{y_p+b} \int_{x_p-b}^{x_p+a} \frac{z^3}{R^5} dxdydz \tag{4}$$

where $E$ is the compression modulus of the foundation soils, $z_1$ and $z_2$ are the buried depth of PE pipeline and compressed soil layer under the action of external stress, respectively.

Based on the Winkler foundation model theory, the foundation reaction on the buried PE pipeline can be defined as,

$$P_1 = kv \tag{5}$$

where $P_1$ is the foundation reaction, $v$ is the vertical displacement of the buried PE pipeline, and $k$ is the coefficient of foundation soils.

The differential equation for pipeline deflection is,

$$E_p I \frac{d^4 v}{dx^4} + Dkv = q_v + k\delta_1(x) \tag{6}$$

where $E_p I$ and $D$ are the bending stiffness and cross-section width of the buried PE pipeline, respectively.

The vertical earth pressure on the pipeline is a non-uniform force, so the additional force on the pipeline can be regarded as numerous concentrated forces q. The pipelines within the load influence length range L are segmented, and the concentrated force at any point of the pipeline can be calculated as,

$$E_p I \frac{v_{i+2} - 4v_{i+1} + 6v_i - 4v_{i-1} + v_{i-2}}{c^4} + Dkv_i = q_i \tag{7}$$

The vertical displacement of the buried pipeline can be calculated as,

$$[A]\{v\} = \{p\}$$

(8)

## 2.2 Model verification

To verify the above established model, an indoor model experiment was conducted. The experimental design and sensor layout was shown in Fig 2, the model test apparatus with a size of 6.0 m × 1.4 m × 1.5 m (length × width × height) was made by steel plate. Besides, steel crossbars were installed on model test apparatus to alleviate the displacement of box wall after soils were filled. The stacking load was adopted on the surface of filled soils with a size of 45 cm × 45 cm, the loading equipment was used an electro-hydraulic servo-hydraulic loading device provided by M.T.S. Industrial Systems Inc., and the max dynamic load was 60 kN/m² in the model verification.

Besides, the buried PE pipeline with a diameter of 90 mm and wall thickness of 9.5 mm was made by Sichuan Rongsu Pipe Industry Co. Two kinds of buried depth were designed, i.e., 0.3 m and 0.8 m. The filled soils were taken from Chengdu, Sichuan Province, it was compacted in layers with a thickness of 10 cm. Additionally, a deformation sensor, which was away from 0.4 m from the loading center axis and located on the top side of buried PE pipeline, was adopted to measure the displacement of pipeline in stacking loading process (Fig 2). The test was repeated three times to ensure data reproducibility. A LVDT-type displacement transducer with a measurement accuracy of ±0.1 mm was employed as the deformation sensor. The results showed a maximum deviation of 2.3% among the three trials, primarily attributed to slight variations in soil compaction. Nevertheless, the load-displacement curves exhibited highly consistent trends, confirming the reliability of the model.

Fig 3 presents a comparative analysis between the experimentally observed and theoretically calculated deformations of PE pipelines buried at varying depths. Notably, the calculated deformations of the buried PE pipelines align closely with the experimental values, particularly for pipelines with a buried depth of 0.8 m. This congruence underscored the accuracy and reliability of the computational models employed in predicting the deformation behavior of buried PE pipelines under

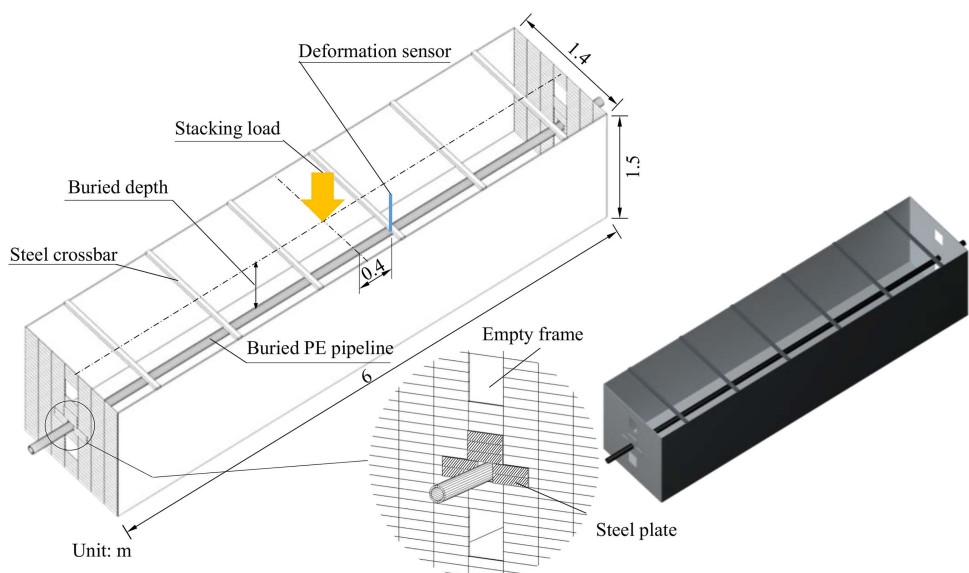

**Fig 2. Experimental design and sensor layout.**

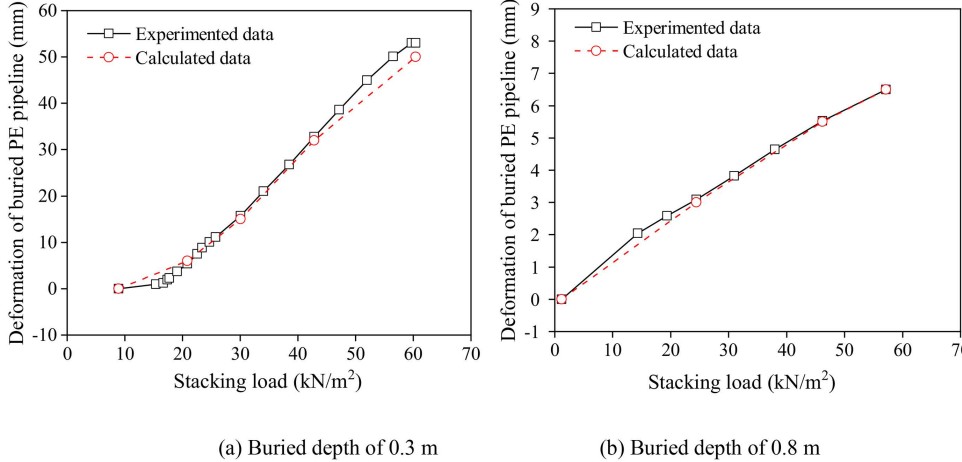

(a) Buried depth of 0.3 m          (b) Buried depth of 0.8 m

**Fig 3. Comparisons between the experimented and calculated deformation of PE pipeline with different buried depths.** (a) Buried depth of 0.3 m (b) Buried depth of 0.8 m.

different conditions (Fig 3b), suggesting that the established buried PE pipeline-soil interaction model can well reflect the variation deformation of buried PE pipeline exposed to stacking load.

### 2.3 Numerical simulation of the working conditions

The numerical simulation of the working conditions for buried PE pipelines is listed in Table 1. Eight types of PE pipeline are used, e.g., DN32, DN40, DN50, DN63, DN75, DN90, DN110, and DN160. Seven level stacking loads (1.0, 1.5, 2.0, 2.5, 3.0, 5.0, and 7.0 kN/m²) and five buried depths of PE pipelines (0.6, 0.8, 1.0, 1.5, and 2.0 m) are designed (Table 1). The stacking load is applied on the surface of soils and transmitted to the buried PE pipeline through the interaction between the pipeline and soils. The basic physical and mechanical parameters of the PE pipeline and soils are listed in Tables 2 and 3, respectively.

## 3 Results and analysis

### 3.1 Displacement of buried PE pipeline

**3.1.1 Variation in displacement with horizontal position.** To explore the deformation variation of buried PE pipeline exposed to stacking load, taking DN50 PE pipeline exposed to 5.0 kN/m² stacking load as an example, the deformation curves of pipelines with various buried depths was shown in Fig 4. The maximum displacement was extracted from a specific point on the horizontal position of the buried PE pipeline, it was observed that the maximum displacement

**Table 1. Design of working conditions for numerical simulation.**

| Type of pipeline | Stacking Load (kN/m²) | Buried depth (m) |
|---|---|---|
| DN32, DN40, DN50, DN63, DN75, DN90, DN110, DN160 | 1.0, 1.5, 2.0, 2.5, 3.0, 5.0, 7.0 | 0.6, 0.8, 1.0, 1.5, 2.0 |

**Table 2. Basic physical and mechanical parameters of PE pipeline.**

| Parameter | Elasticity modulus (MPa) | Poisson's Ratio | Density (kg/m³) | Yield stress (MPa) |
|---|---|---|---|---|
| Value | 1115 | 0.45 | 951 | 15.4 |

**Table 3. Physical and mechanical parameters of soils.**

| Parameter | Density (kg/m³) | Elasticity modulus (MPa) | Poisson's ratio | Internal friction angle (°) | Cohesive Strength (kPa) |
|---|---|---|---|---|---|
| Value | 1867 | 20000 | 0.4 | 18.4 | 29.3 |

occurred at the center of the buried PE pipeline because the stacking load was applied on the soils above the center of the pipeline. Moreover, the displacements of the pipeline, including upper and lower deformations, as well as diameter change, decreased with the increase of distance from the center of the pipeline, particularly for the upper and lower deformations. Specifically, within a 2.0 m distance from the center of the pipeline, the upper deformation was larger than the lower deformation at the same horizontal position since the stacking load transferred to the upper part of the pipeline was larger than the load on the lower part, with both ends of the pipeline being fixed. However, beyond a 3.0 m range from the center of the pipeline, the displacements at the upper and lower parts of the pipeline were nearly identical. Furthermore, as the buried depth increased, the difference between the upper and lower deformations of the pipeline decreased (Fig 4). This might be attributed to the fact that the dissipation effect of the filling soils on stacking loads became more significant with buried depth increased.

**3.1.2 Relationship between the displacement and stacking load.** Fig 5 displays the maximum deformation variation of buried DN50 PE pipeline with different stacking loads. The maximum displacement of buried PE pipeline increased with the stacking load raised. Because for the pipeline buried at the same depth, the load acted on the pipeline was positively related to the stacking load on the surface of filling soils. Besides, it also obtained that the relationship between the maximum displacement and stacking load can be expressed as,

$$y = a + bp \tag{9}$$

where $y$ is the maximum displacement of buried PE pipeline, $a$ and $b$ are the fitted parameters, $p$ is the stacking load acted on the surface of filling soils.

In practical applications, the deformation of buried PE pipelines is influenced by a multitude of factors. These include the physical properties of the surrounding soils, such as particle size distribution and compactness, as well as pipeline-specific parameters, load types and magnitudes, and various geological and environmental conditions. Among these, the buried load emerges as a crucial factor influencing pipeline deformation, with the load on the pipeline typically increasing with the addition of overlying weight. Furthermore, the dissipation of overlying load by the soils also exerts an impact on the mechanical properties of the pipeline, thereby influencing its deformation behavior (Fig 5).

**3.1.3 Relationship between the displacement and buried depth.** Fig 6 illustrates the variation in deformation exhibited by buried DN50 PE pipelines across varying buried depths. Notably, irrespective of the stacking load applied, the deformation of these buried PE pipelines diminished progressively with an increase in buried depth, accompanied by a corresponding decrease in the rate of deformation reduction. Through fitting the correlation using different mathematical formulations revealed that a power function aptly captured the relationship between the deformation and the buried depth, offering a quantitative understanding of this phenomenon,

$$y = c \cdot d_p{}^d \tag{10}$$

where $c$ and $d$ are the fitted parameters, $d_p$ is the buried depth acted in the surface of filling soils.

Table 4 listed the fitting parameters of the relationship between the PE pipeline and deformation under various stacking loads and PE pipelines. It can be found that the maximum deformation of PE pipeline presented a positive correlation with the stacking load and a negative correlation with the buried depth. The high correlation coefficient ($R^2 > 0.90$) exhibited that the model (Eq. 10) could well predict the maximum deformation of buried PE pipeline exposed to various stacking

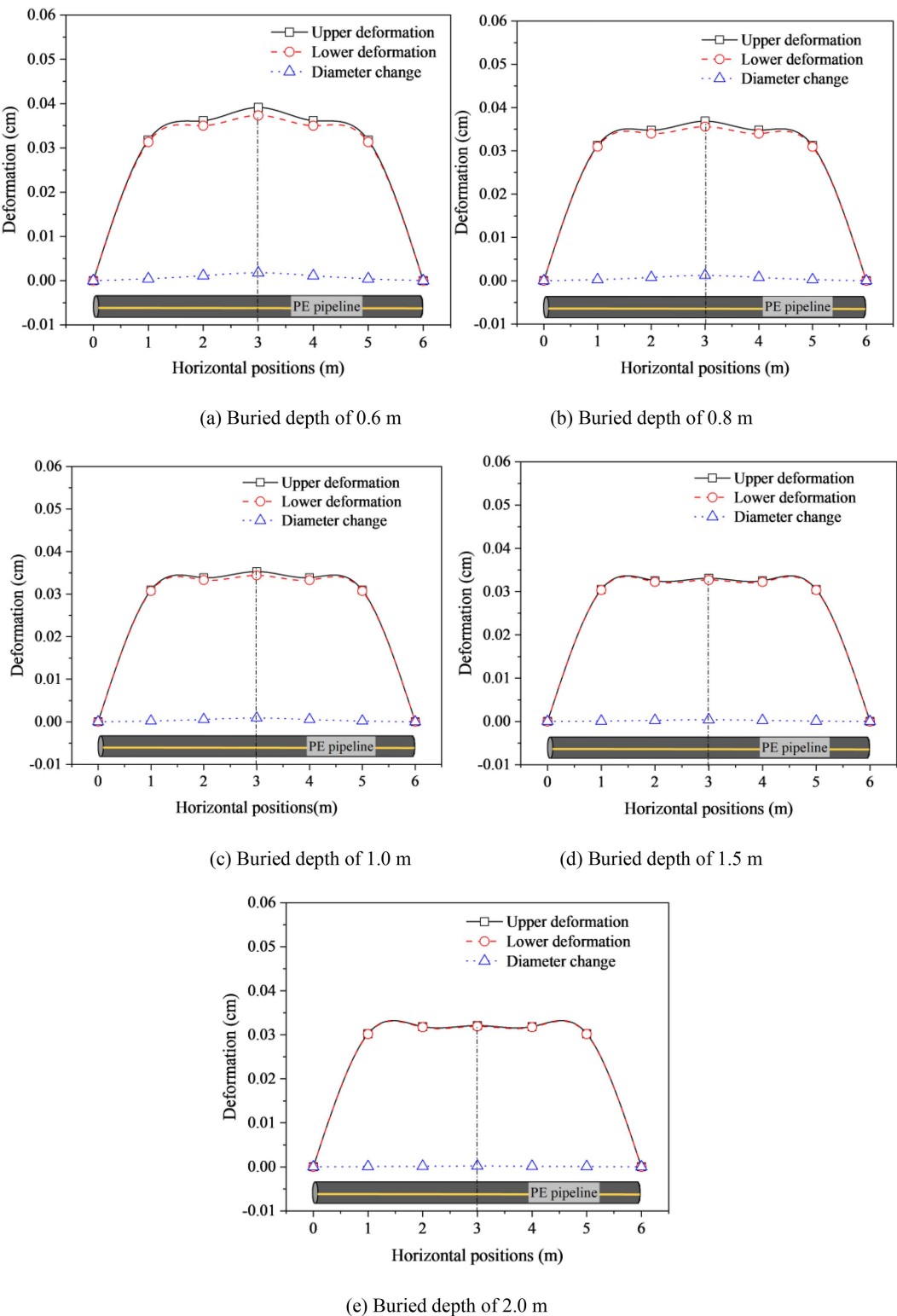

(a) Buried depth of 0.6 m

(b) Buried depth of 0.8 m

(c) Buried depth of 1.0 m

(d) Buried depth of 1.5 m

(e) Buried depth of 2.0 m

**Fig 4. Deformation of buried DN50 PE pipeline exposed to 5000 kN/m² stacking load with various buried depths.** (a) Buried depth of 0.6 m (b) Buried depth of 0.8 m (c) Buried depth of 1.0 m (d) Buried depth of 1.5 m (e) Buried depth of 2.0 m.

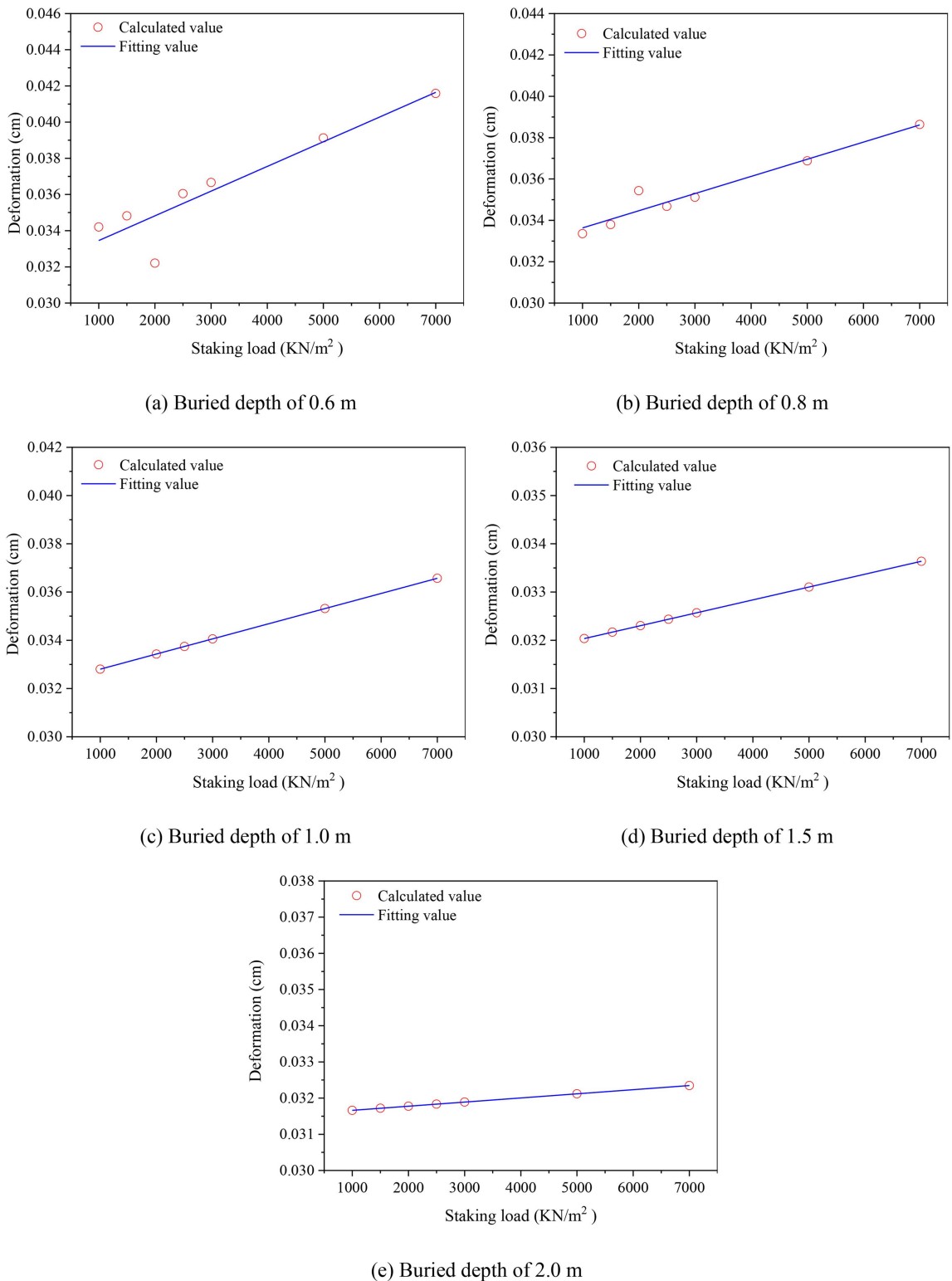

(a) Buried depth of 0.6 m

(b) Buried depth of 0.8 m

(c) Buried depth of 1.0 m

(d) Buried depth of 1.5 m

(e) Buried depth of 2.0 m

**Fig 5. Maximum deformation variation of buried DN50 PE pipeline with different stacking loads.** (a) Buried depth of 0.6 m (b) Buried depth of 0.8 m (c) Buried depth of 1.0 m (d) Buried depth of 1.5 m (e) Buried depth of 2.0 m.

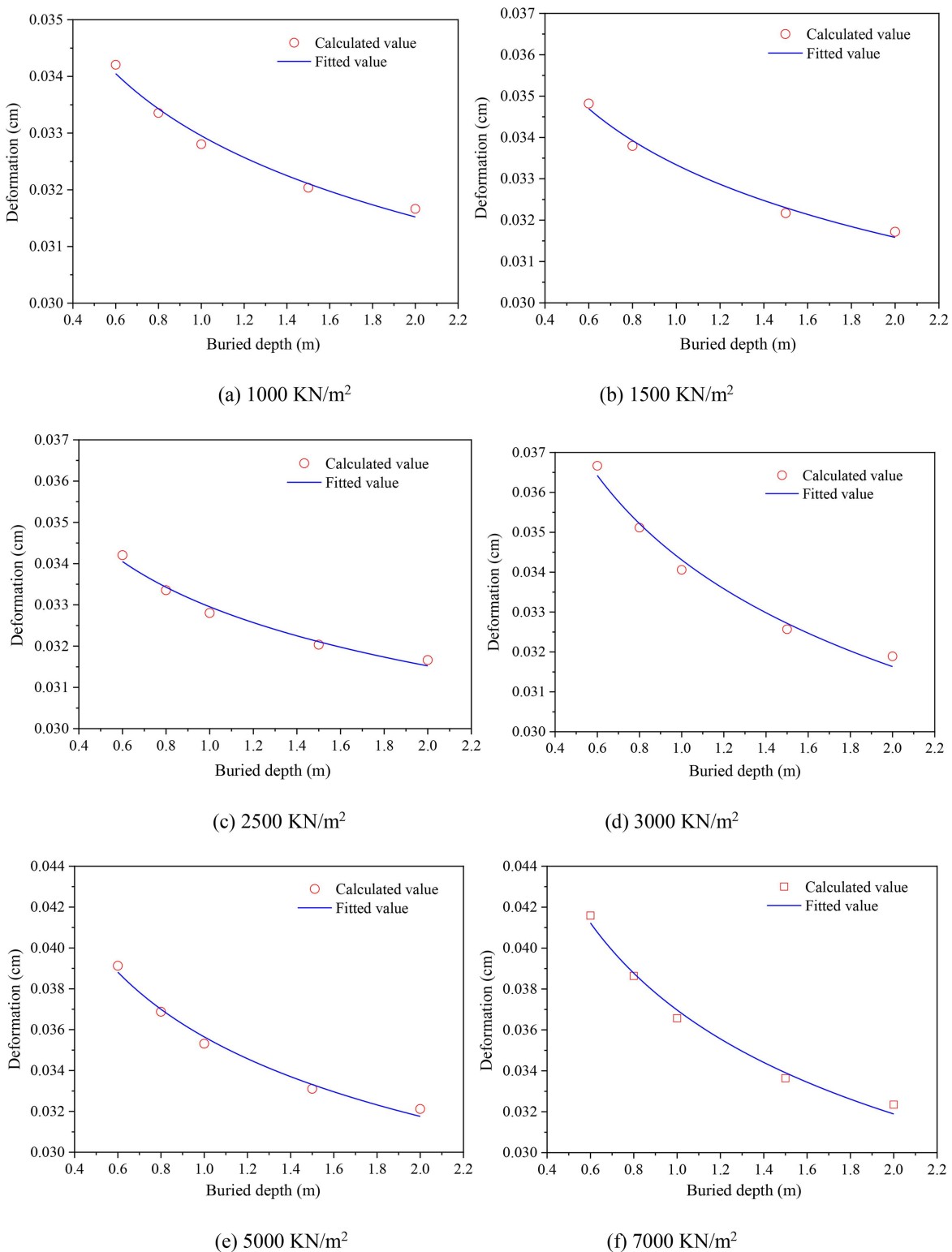

**Fig 6. Deformation variation of buried DN50 PE pipeline with different buried depths.** (a) 1000 kN/m² (b) 1500 kN/m² (c) 2500 kN/m² (d) 3000 kKN/m²(e) 5000 kN/m² (f) 7000 kN/m².

**Table 4. Fitting parameters of the relationship between the PE pipeline and deformation under various stacking loads and PE pipelines.**

| Type of pipeline | Stacking load (kN/m$^2$) | c | d | R$^2$ |
|---|---|---|---|---|
| DN32 | 1000 | 0.0330 | −0.0634 | 0.9822 |
| | 1500 | 0.0333 | −0.0767 | 0.9830 |
| | 2000 | 0.0337 | −0.0897 | 0.9837 |
| | 2500 | 0.0340 | −0.1042 | 0.9788 |
| | 3000 | 0.0343 | −0.1152 | 0.9849 |
| | 5000 | 0.0356 | −0.1637 | 0.9870 |
| | 7000 | 0.0366 | −0.1984 | 0.9571 |
| DN40 | 1000 | 0.0330 | −0.0633 | 0.9806 |
| | 1500 | 0.0333 | −0.0767 | 0.9820 |
| | 2000 | 0.0336 | −0.0899 | 0.9830 |
| | 2500 | 0.0340 | −0.1029 | 0.9839 |
| | 3000 | 0.0343 | −0.116 | 0.9847 |
| | 5000 | 0.0356 | −0.1637 | 0.9870 |
| | 7000 | 0.0369 | −0.2105 | 0.9890 |
| DN50 | 1000 | 0.0330 | −0.0641 | 0.9815 |
| | 1500 | 0.0333 | −0.0792 | 0.9915 |
| | 2000 | 0.0341 | −0.1137 | 0.9005 |
| | 2500 | 0.0340 | −0.1042 | 0.9845 |
| | 3000 | 0.0343 | −0.1171 | 0.9852 |
| | 5000 | 0.0357 | −0.1668 | 0.9876 |
| | 7000 | 0.0370 | −0.2133 | 0.9894 |
| DN63 | 1000 | 0.0329 | −0.0653 | 0.9154 |
| | 1500 | 0.0332 | −0.0853 | 0.9546 |
| | 2000 | 0.0336 | −0.0953 | 0.97911 |
| | 2500 | 0.0334 | −0.1072 | 0.9840 |
| | 3000 | 0.0343 | −0.1204 | 0.9848 |
| | 5000 | 0.0357 | −0.1759 | 0.9923 |
| | 7000 | 0.0371 | −0.2187 | 0.9893 |
| DN75 | 1000 | 0.0330 | −0.0669 | 0.9812 |
| | 1500 | 0.0333 | −0.0811 | 0.9825 |
| | 2000 | 0.0336 | −0.0950 | 0.9836 |
| | 2500 | 0.0340 | −0.1088 | 0.9844 |
| | 3000 | 0.0343 | −0.1222 | 0.9852 |
| | 5000 | 0.0358 | −0.1754 | 0.9872 |
| | 7000 | 0.0371 | −0.2222 | 0.9896 |
| DN90 | 1000 | 0.0329 | −0.0678 | 0.9813 |
| | 1500 | 0.0329 | −0.0678 | 0.9813 |
| | 2000 | 0.0328 | −0.0559 | 0.9593 |
| | 2500 | 0.0340 | −0.1106 | 0.9845 |
| | 3000 | 0.0343 | −0.1258 | 0.9851 |
| | 5000 | 0.03580 | −0.1787 | 0.9873 |
| | 7000 | 0.03780 | −0.2316 | 0.9666 |

*(Continued)*

**Table 4.** (Continued)

| Type of pipeline | Stacking load (kN/m²) | c | d | R² |
|---|---|---|---|---|
| DN110 | 1000 | 0.0328 | −0.0689 | 0.9775 |
| | 1500 | 0.0332 | −0.0835 | 0.9795 |
| | 2000 | 0.0336 | −0.0980 | 0.9811 |
| | 2500 | 0.0340 | −0.1121 | 0.9823 |
| | 3000 | 0.0343 | −0.1260 | 0.9834 |
| | 5000 | 0.0358 | −0.1792 | 0.9867 |
| | 7000 | 0.03730 | −0.2288 | 0.9890 |
| DN160 | 1000 | 0.0326 | −0.0739 | 0.9799 |
| | 1500 | 0.0330 | −0.0902 | 0.9814 |
| | 2000 | 0.0333 | −0.1181 | 0.9914 |
| | 2500 | 0.0338 | −0.1217 | 0.9836 |
| | 3000 | 0.0342 | −0.1370 | 0.9844 |
| | 5000 | 0.0358 | −0.1953 | 0.9872 |
| | 7000 | 0.0373 | −0.2493 | 0.9894 |

loads and buried depths (Table 4). Furthermore, it could also be preliminary concluded that with the increase of buried depth, the maximum deformation of PE pipeline gradually decreased, which was because the overlying soil dispersed the accumulation load, thus reduced the bearing pressure of PE pipeline and decreased its maximum deformation. The above results indicated that the displacement of the PE pipeline was positively correlated with the magnitude of the stacking load, which laied a foundation for further research on its stress variation trends.

## 3.2 Stress of buried PE pipeline

**3.2.1 Variation stress with horizontal position.** Building upon the deformation characteristics analyzed in Section 3.1, this section further investigates the influence of soil-pipeline interaction on the stress distribution patterns of the pipeline. To elucidate the stress variations in a buried PE pipeline subjected to various stacking loads and buried depths, Fig 7 presents a case study focusing on a DN32 pipeline buried at a depth of 0.8 m, specifically detailing how the stress varies along the horizontal position. Generally, the stress experienced by the buried PE pipeline was primarily influenced by two factors, i.e., the weight of the surrounding soil and the surface stacking loads. Typically, the surface stacking load transmited its force downwards at θ angles (known as stress diffusion angles), with the impact being significantly more pronounced within the inside θ angles compared to the outside. The stress of buried PE pipeline caused by soil gravity could be calculated as,

$$\sigma_c = \sum_{i=1}^{n} r_i h_i$$

(11)

where $n$ is the number of soil layers, $r_i$ and $h_i$ are the bulk density and thickness of the soils.

Fig 7 clearly demonstrated that the peak stress in the buried PE pipeline occurred precisely at the center of the load coverage area. As the stacking loads increased, the stress distribution curve along the pipeline gradually flattened out. Notably, the stress within the pipeline intensified with the escalation of stacking loads, particularly pronounced at the horizontal center of the pipeline. Specifically, the maximum stress values for the PE pipeline subjected to stacking loads of 1000, 1500, 2000, 2500, 3000, 5000, and 7000 N/m² were 107.58, 270.64, 361.28, 452.00, 609.67, 1092.34, and 1459.64 Pa, respectively, highlighting a direct correlation between the stacking load and pipeline stress. Besides, the maximum

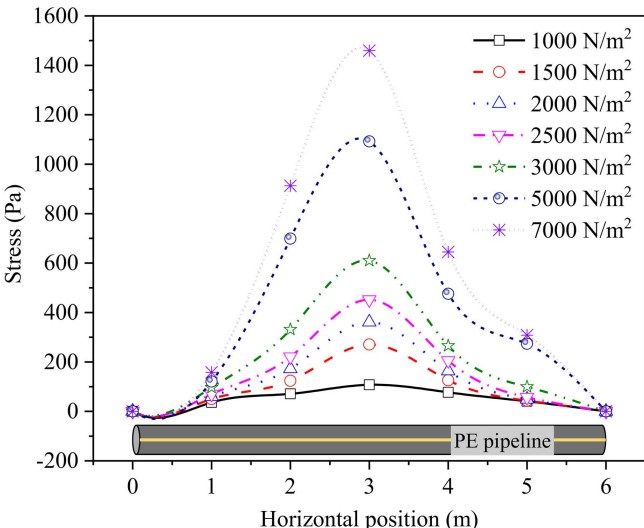

**Fig 7. Stress of DN32 PE pipeline with the buried depth of 0.8 m.**

stress of the buried PE pipeline accounted for 10.76, 18.04, 18.06, 18.08, 20.32, 21.85, and 20.85% of the corresponding stacking loads due to the dissipation effect of filled soils on the stacking loads.

Fig 8 shows the stress distribution of DN32 PE pipeline buried at the depth of 0.8 m with different stacking loads. Both the upper and lower stresses of the pipeline increased as the distance from the pipeline's horizontal center (located at 3.0 m) decreased. For pipelines subjected to relatively low stacking loads, such as 1500 N/m² and 2000 N/m², the upper stress exceeded the lower stress within the horizontal position range of 2.0 to 4.0 m. Conversely, within the ranges of 0 to 2.0 m and 4.0 to 6.0 m, the difference between the upper and lower stresses was minimal (as depicted in Fig 8a and 8b). Specifically, for the pipelines exposed to 1500 N/m² and 2000 N/m² stacking loads, the maximum differences in stress between the upper and lower surfaces were 45.40 Pa and 77.53 Pa, respectively (Fig 8a and 8b). Those might from the fact that under the small stacking loads, the stress transferred to the pipeline was small due to soil dissipation, resulting in small pipeline deformation. Thus, the upper stress was higher than the lower stress (Fig 8a and 8b). However, for the buried PE pipeline exposed to large stacking loads (e.g., 5000 N/m² and 7000 N/m²), the upper stress was larger than the lower stress in the horizon position range of 2.5~3.5m, while reversed results occurred in the horizon position ranges of 0~2.5m and 3.5~6.0m (Fig 8c and 8d). Because for the large stacking loads, the filled soils would produce a large range of settlement deformation, resulting in the dislocation of the settlement section of the soils. Thus, the stress of buried pipeline increased. Generally, the stress change was mainly caused by the bending deformation in the center of the pipeline and the stretching deformation of the whole pipeline. The stretching deformation would offset the compressive deformation at the top of the pipeline and aggravate the deformation at the bottom of pipeline. Consequently, the upper stress in the center horizon position was larger than the lower stress, while reversed results occurred far away from the center horizon position (Fig 8c and 8d).

**3.2.2 Relationship between the stress and buried depth.** Fig 9 presents the relationship between the stress and buried depth for the various PE pipeline. The variation in stress for buried PE pipelines could be categorized into three distinct stages. Specifically, within a buried depth of 0.5m to 0.75m, the stress underwent a sharp decline as the depth increased. Subsequently, from 0.75m to 1.25m, the stress gradually diminished at a slower pace. Finally, for buried depths ranging from 1.25m to 2.0m, the stress decreased slightly and tended to stabilize as the depth further increased (Fig 9). This phenomenon could be attributed to the soil surrounding the buried PE pipeline

 

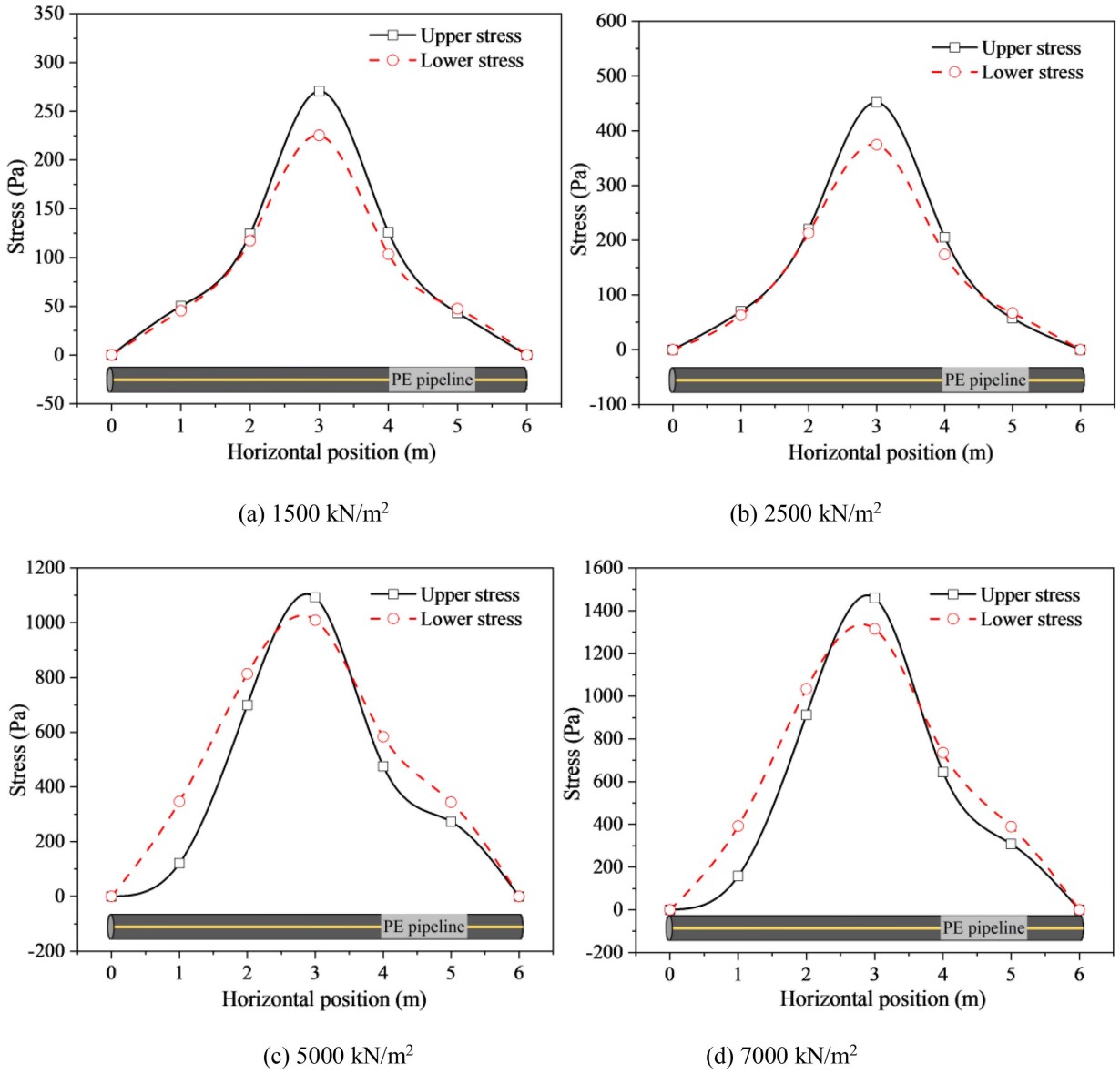

**Fig 8. Stress of DN32 PE pipeline buried at the depth of 0.8 m with different stacking loads.** (a) 1500 kN/m² (b) 2500 kN/m² (c) 5000 kN/m² (d) 7000 kN/m².

effectively mitigating the load. Furthermore, Fig 9 also revealed that as the pipeline diameter increased, the peak stress of the buried PE pipeline gradually decreased. This was because the thicker walls of larger diameter pipelines effectively distributed the accumulated loads, with the enhanced stiffness of these thicker walls contributing to the reduction in peak stress.

## 4. Conclusions

This study investigated the variations in deformation and stress of buried PE pipelines via numerical simulations, and some conclusions were drawn,

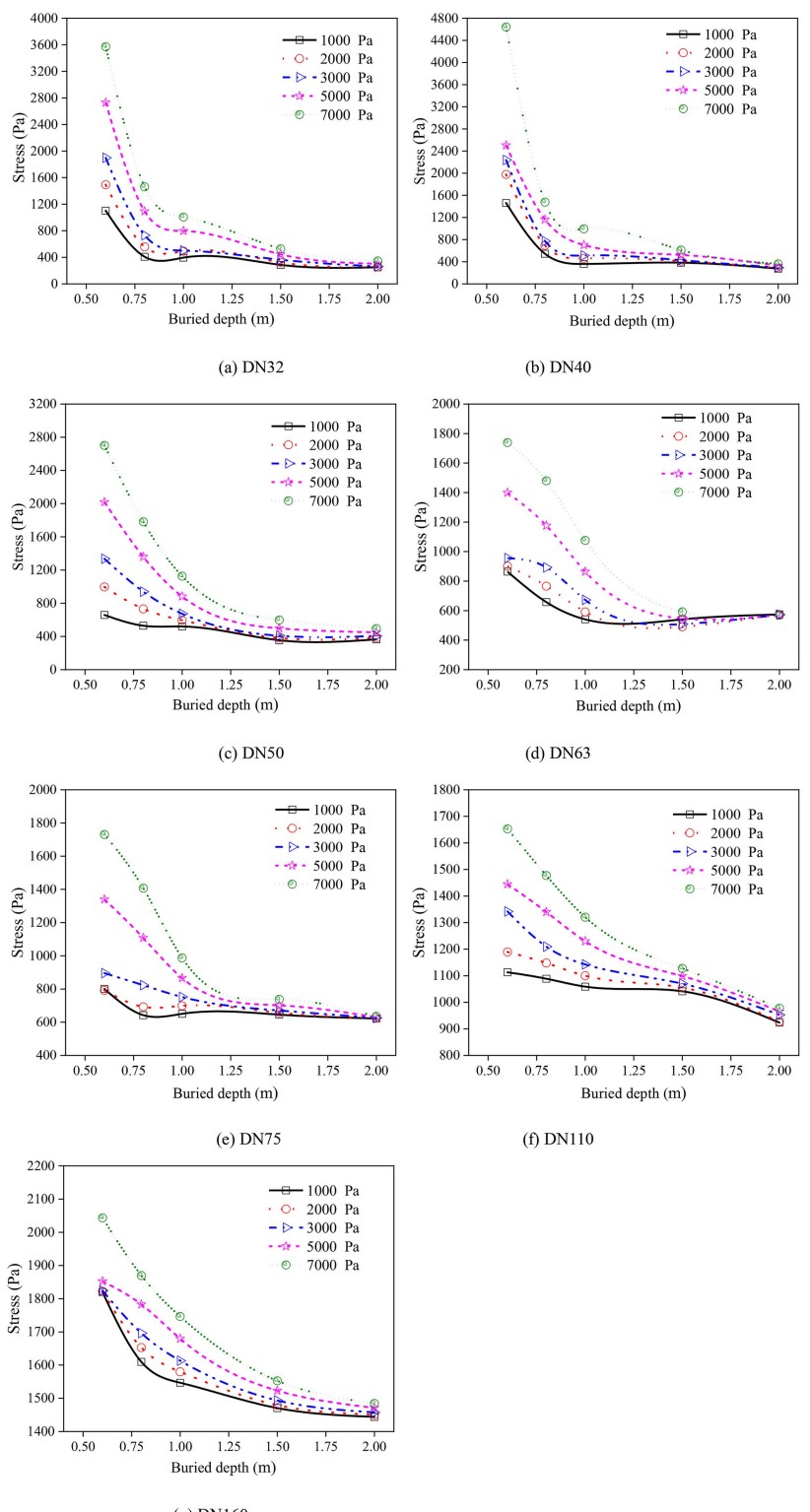

**Fig 9. Relationship between the stress and buried depth for the PE pipeline.** (a) DN32 (b) DN40 (c) DN50 (d) DN63 (e) DN75 (f) DN110 (g) DN160.

(1)  The maximum deformation of buried PE pipeline under stacking loads occurred at the central position. Besides, the upper and lower deformations, and diameter change decreased with the increase of distance from the pipeline center, particularly for the upper and lower deformations. Additionally, the disparity between the upper and lower deformations progressively diminished with buried depth.

(2) A linear correlation existed between the applied stacking load and maximum pipeline deformation. The deformation of buried PE pipeline decreased with the increase of buried depth, as well the rate of deformation reduction. Besides, the deformation-depth relationship could be used to describe by a power function.

(3) Peak stress concentrations in buried PE pipelines were localized at the load center. The stress distribution profile transitioned from peaked to flattened with increasing stacking loads. Besides, the stress of pipeline increased with the stacking load raised.

(4) For the buried PE pipeline exposed to small stacking loads, the upper stress exceeded the lower stress within the horizontal range of 2.0~4.0 m, while there were little gaps between the upper and lower stresses in the horizon position ranges of 0~2.0 and 4.0~6.0 m. Under large stacking loads, the upper stress was higher than the lower stress in the horizon position range of 2.5~3.5m, while reversed results occurred in the horizon position ranges of 0~2.5m and 3.5~6.0m. Furthermore, the peak stress of buried PE pipelines decreased progressively with pipe diameter increased.

## Supporting information

**S1 File.  Supplementary data files.**
(XLSX)

## Author contributions

**Conceptualization:** Pengcheng Li.

**Data curation:** Fei Wang.

**Formal analysis:** Dong Lin, Ming Wen.

**Investigation:** Pengcheng Li.

**Software:** Ming Wen.

**Supervision:** Jianguo Lu.

**Validation:** Pengcheng Li, Dong Lin, Chang Liu.

**Writing – original draft:** Pengcheng Li.

**Writing – review & editing:** Dong Lin, Fei Wang, Chang Liu, Mingnan Sun, Jianguo Lu.

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
