## [Decision Letter · Decision Letter 0]

15 Apr 2025

Thank you for submitting your manuscript to PLOS ONE. After careful consideration, we feel that it has merit but does not fully meet PLOS ONE’s publication criteria as it currently stands. Therefore, we invite you to submit a revised version of the manuscript that addresses the points raised during the review process.

We look forward to receiving your revised manuscript.

Kind regards,

Dr. Guojin Qin

Academic Editor

PLOS ONE

“This research was supported by the Science and Technology Project of PetroChina Southwest Oil and Gas Field Company (Grant No. 20220307-10), and the Natural Science Foundation of Sichuan Province (Grant No. 2022NSFSC0429).”

Reviewers' comments:

Reviewer's Responses to Questions

**Comments to the Author**

1. Is the manuscript technically sound, and do the data support the conclusions?

Reviewer #1: Yes

Reviewer #2: Yes

2. Has the statistical analysis been performed appropriately and rigorously?

Reviewer #1: Yes

Reviewer #2: Yes

3. Have the authors made all data underlying the findings in their manuscript fully available?

Reviewer #1: Yes

Reviewer #2: Yes

4. Is the manuscript presented in an intelligible fashion and written in standard English?

Reviewer #1: Yes

Reviewer #2: Yes

Reviewer #1: This study presents a numerical investigation into the stress and deformation behavior of buried polyethylene (PE) pipelines subjected to stacking loads. The study develops a coupled soil–pipeline interaction model, validated by laboratory-scale experiments, and explores the effects of various pipeline diameters, burial depths, and loading conditions on mechanical responses. The findings contribute valuable insights into the structural safety and design optimization of buried pipeline systems under surface loading. Overall, the manuscript is logically structured, the results are systematically analyzed. However, certain issues remain, including unclear justification for parameter selection, insufficient methodological detail in the experimental setup. These aspects require revision and clarification to further improve the scientific rigor and practical relevance of the paper.Redundant phrasing in the Abstract: "buried PE pipeline exposed to stacking load", This phrase is repeated multiple times in the abstract, which affects readability. Rephrase to avoid redundancy. In the introduction, It provides a broad review of existing work but does not explicitly clarify the innovation or research gap addressed by this study. Please Add a paragraph at the end of the Introduction stating the gap in previous literature and how your study contributes (e.g., "Few studies have systematically investigated the stress-deformation coupling in buried PE pipelines under varying stacking loads and burial depths. This paper develops...").In Section 2.2, the model validation section does not mention the number of repetitions, sensor resolution, or error margins.Please add the details such as the number of repeated tests, precision of the displacement sensor (e.g., ±0.1 mm), and discuss experimental variability to enhance credibility.The terms “pipe” and “pipeline” are used interchangeably throughout the manuscript. It is recommended to standardize the terminology for clarity and consistency.In Equation (3), several symbols such as “τ” are not clearly defined in the text. Please ensure that all variables and parameters (e.g., τ, R, v) are explicitly introduced and described when each equation is presented, to improve clarity and mathematical rigor.In Equation (11), the variable “ri” should be formatted in italic with the “i” as a subscript. Please review and correct the formatting of all equations to ensure consistency with standard mathematical typesetting.In Section 3, although the content is rich and well-analyzed, the transitions between subsections are not sufficiently smooth. It is recommended to include brief concluding or transitional sentences at the end of each subsection to improve coherence and guide the reader more effectively through the progression of the analysis.In Table 3, the unit notation for stacking load appears to be incorrect or inconsistent. Please review and correct the unit if necessary. Additionally, it is recommended to check the entire manuscript to ensure that all physical units are accurately and consistently presented.

Reviewer #2: In this paper, the stress and deformation behavior of buried PE pipeline exposed to stacking load were studied. The research has some novelty. But there are still parts that need to be corrected before acceptance.

1) The title should use "deformation" instead of the misspelled "defo-mation".

2) The variables in the formulas should be explained and defined upon their first occurrence, with consistent formatting throughout.

3) The terms "pipe" and "pipeline" in the text should be revised and standardized to one consistent term.

4) Lines 189-190: The model test box dimensions are stated as height 1.4 m and width 1.5 m, but Figure 2 shows height 1.5 m and width 1.4 m.

5) The coordinate format in Figure 5 needs to be unified, and the units in Figure 8 require superscript formatting. A full review of the manuscript for consistency is recommended.

6) Line 214: The reference to "Fig. 2b" is invalid as no such figure exists in the manuscript. Please confirm whether this subfigure is missing or requires renumbering.

7) Lines 316-317: The conclusion about correlation should be explicitly supported by data analysis. Please clarify how the correlation was derived.

8) The 7000 Pa curve in Figure 9 is unclear and requires modification for better visibility.

9) The Conclusions section must be revised to align with the updated manuscript. Focus on highlighting the core contributions of the study and ensure the description is concise and focused.

**Do you want your identity to be public for this peer review?** For information about this choice, including consent withdrawal, please see our Privacy Policy

Reviewer #1: No

Reviewer #2: No

---

## [Author Response · Author response to Decision Letter 1]

30 Jun 2025

We much appreciate the constructive and insightful suggestions made by the editor and reviewers. We have addressed all of the comments, and list below each comment and our response (in Response to Reviewers).

---

## [Decision Letter · Decision Letter 1]

18 Jul 2025

Numerical study on the stress and deformation behavior of buried PE pipeline exposed to stacking load

PONE-D-25-17420R1

Dear Dr. Lu,

We’re pleased to inform you that your manuscript has been judged scientifically suitable for publication and will be formally accepted for publication once it meets all outstanding technical requirements.

Kind regards,

Dr. Guojin Qin

Academic Editor

PLOS ONE

Additional Editor Comments (optional):

Reviewers' comments:

Reviewer's Responses to Questions

**Comments to the Author**

Reviewer #1: All comments have been addressed

Reviewer #2: All comments have been addressed

2. Is the manuscript technically sound, and do the data support the conclusions?

Reviewer #1: Yes

Reviewer #2: Yes

3. Has the statistical analysis been performed appropriately and rigorously?

Reviewer #1: (No Response)

Reviewer #2: Yes

4. Have the authors made all data underlying the findings in their manuscript fully available?

Reviewer #1: Yes

Reviewer #2: Yes

5. Is the manuscript presented in an intelligible fashion and written in standard English?

Reviewer #1: Yes

Reviewer #2: Yes

Reviewer #1: (No Response)

Reviewer #2: (No Response)

**Do you want your identity to be public for this peer review?** For information about this choice, including consent withdrawal, please see our Privacy Policy

Reviewer #1: No

Reviewer #2: No

---

## [Editor Report · Acceptance letter]

PONE-D-25-17420R1

PLOS ONE

Dear Dr. Lu,

I'm pleased to inform you that your manuscript has been deemed suitable for publication in PLOS ONE. Congratulations! Your manuscript is now being handed over to our production team.

Kind regards,

on behalf of

Dr. Guojin Qin

Academic Editor

PLOS ONE